# Design of an Integrated Platform for Mapping Residential Exposure to Rf-Emf Sources

**DOI:** 10.3390/ijerph17155339

**Published:** 2020-07-24

**Authors:** Corentin Regrain, Julien Caudeville, René de Seze, Mohammed Guedda, Amirreza Chobineh, Philippe de Doncker, Luca Petrillo, Emma Chiaramello, Marta Parazzini, Wout Joseph, Sam Aerts, Anke Huss, Joe Wiart

**Affiliations:** 1Institut National de l’Environnement Industriel et des Risques (INERIS), Parc, 60550 Verneuil en Halatte, France; julien.caudeville@ineris.fr (J.C.); rene.de-seze@ineris.fr (R.d.S.); 2LAMFA, UMR CNRS 7352, Université de Picardie Jules Verne, 33 rue Saint-Leu, 80039 Amiens, France; mohamed.guedda@u-picardie.fr; 3PériTox, UMR_I 01, CURS, Université de Picardie Jules Verne, 80025 Amiens, France; 4LTCI Telecom Paris, Chaire C2m, Institut Polytechnique de Paris, 91120 Palaiseau, France; amirreza.chobineh@telecom-paris.fr (A.C.); joe.wiart@telecom-paris.fr (J.W.); 5OPERA—Wireless Communications Group, Université Libre de Bruxelles, 1050 Brussels, Belgium; pdedonck@ulb.ac.be (P.d.D.); lpetrill@ulb.ac.be (L.P.); 6CNR IEIIT—Consiglio Nazionale delle Ricerche, Istituto di Elettronica e di Ingegneria dell’Informazione e delle Telecomunicazioni, 20133 Milan, Italy; emma.chiaramello@ieiit.cnr.it (E.C.); marta.parazzini@ieiit.cnr.it (M.P.); 7Department of Information Technology, Ghent University, 9052 Ghent, Belgium; Wout.joseph@ugent.be (W.J.); sam.aerts@ugent.be (S.A.); 8Institute for Risk Assessment Sciences, Utrecht University, 3508 Utrecht, The Netherlands; a.huss@uu.nl

**Keywords:** radiofrequency electromagnetic fields, spatiotemporal exposure assessment, data fusion, Monte Carlo approach

## Abstract

Nowadays, information and communication technologies (mobile phones, connected objects) strongly occupy our daily life. The increasing use of these technologies and the complexity of network infrastructures raise issues about radiofrequency electromagnetic fields (Rf-Emf) exposure. Most previous studies have assessed individual exposure to Rf-Emf, and the next level is to assess populational exposure. In our study, we designed a statistical tool for Rf-Emf populational exposure assessment and mapping. This tool integrates geographic databases and surrogate models to characterize spatiotemporal exposure from outdoor sources, indoor sources, and mobile phones. A case study was conducted on a 100 × 100 m grid covering the 14th district of Paris to illustrate the functionalities of the tool. Whole-body specific absorption rate (SAR) values are 2.7 times higher than those for the whole brain. The mapping of whole-body and whole-brain SAR values shows a dichotomy between built-up and non-built-up areas, with the former displaying higher values. Maximum SAR values do not exceed 3.5 and 3.9 mW/kg for the whole body and the whole brain, respectively, thus they are significantly below International Commission on Non-Ionizing Radiation Protection (ICNIRP) recommendations. Indoor sources are the main contributor to populational exposure, followed by outdoor sources and mobile phones, which generally represents less than 1% of total exposure.

## 1. Introduction

Nowadays, wireless communication devices and connected objects are part of our personal and professional daily life. In 2018, number of worldwide public Wi-Fi hotspots reached 168.6 million. The same year, the number of mobile devices and connections reached 8.8 billion [1]. With the advent of the Internet-of-Things (IoT), billions of connected objects are also being deployed [2]. Both evolutions have a major impact on our information and communications technology (ICT) usage and on our economy. By 2023, the number of worldwide mobile devices will increase to 13.1 billion, with 4G and 5G, respectively, generating 46% and 11% of global connections. With the increase of smartphones, machine-to-machine (M2M) connections, connected TVs, and tablets, the number of devices will reach 3.6 per capita [1]. Over the last two decades, technologies and networks have evolved from GSM (Global System for Mobile Communication) to UMTS (Universal Mobile Telecommunications System; 3G) and LTE (Long Term Evolution; 4G). Wireless networks now have variable densities and use different technologies. Networks are no longer just composed of macro cells with antennas on roofs or micro cells with antennas in streets. They are also composed of outdoor small cells with antennas on lampposts or bus stops and of indoor small cells with antennas in corridors as well as femtocells and Wi-Fi hotspots. Network complexity will furthermore increase with the advent of self-organizing networks and offloading, i.e., information provided through different networks [3]. The future 5G technologies, including IoT and Low Power Wide Area (LPWA) networks, will increase this network complexity. The new technologies allowing the IoT and the pervasive use of wireless ICT have emphasized the need for characterizing the electromagnetic field (Emf) exposure not only for general public information but also for health researchers or policy-makers and entities in charge of radiofrequency (Rf) exposure monitoring. Individual exposure to Rf-Emf must address both the exposure induced by base stations and access points and the exposure induced by a person’s portable device during use, as well as the exposure induced by surrounding devices.

Some prior efforts have analyzed exposure to Rf-Emf sources. Using dedicated measurement protocols, large measurement campaigns have been organized to respond to public concern about Rf-Emf exposure and its potential health effects [4,5]. Other studies assessed the exposure to Rf-Emf with simulation and modeling tools. For instance, the Samper and B-WARE projects initiated work on the use of geostatistical tools for outdoor exposure [6,7,8]. The Mobi-Expo project analyzed the exposure induced by wireless communication devices using the application “Xmobisense,” developed by C2M (“Modeling, Characterization and Control of exposure to electromagnetic waves” chair). The project studied the uplink exposure through parameters such as call duration and usage (laterality, hands-free kit) [9,10]. However, spatial Rf-Emf exposure, population statistics, and the usage of telecommunication technology (ICT) have not been combined into an integrated assessment. The first project that has analyzed the exposure according to an integrated assessment was the LEXNET project (Low EMF Exposure Networks) [11]. This project defined a new exposure indicator, the Exposure Index (E.I.), able to quantify the individual exposure induced by both the access points, i.e., the downlink exposure and mobile devices, i.e., the uplink exposure. The LEXNET project defined and used the E.I. but did not assess the population Rf-Emf exposure originating from all networks [11,12]. Another study (DICER; “Definition of Indicators for the Characterization of Radiofrequency Exposure”) has initiated work on the development of indicators to characterize exposure [13].

The analysis of the efforts that have been carried out to date shows that they are mainly focused on individual exposure. Few studies have used an integrated approach, taking into account on the one hand Rf-Emf emitted by access points and base stations and, on the other hand, the population distribution, characteristics, and ICT usage. To date, the methods that have been applied are limited with respect to the evolving networks and technologies. This overview of previous studies also shows that the existing Rf-Emf exposure assessment tools do not involve multiple parameters, as is the case in other environmental domains, where platforms such as PLAINE (“Environmental Inequalities Analysis Platform”) allow for a dynamic assessment, related to a geographical area, and responding to multiple determinants [14]. Previous studies have often addressed discreet parts of the problem with different metrics (electric-field strength, in V/m, for the exposure induced by access points, and specific absorption rate, in W/kg, for mobile phones) and different approaches (worst case for compliance vs. mean value for epidemiological studies). Previous studies have used deterministic methods such as propagation tools based on ray tracing techniques that require the exact position of the antennas, which is more and more time consuming given the increasing number of access points such as Wi-Fi hotspots and femtocells. Regarding prior studies, integrating indoor and outdoor exposures in a populational approach to assess the Rf-Emf exposure would give relevant information about population exposure distribution. This approach must characterize the complexity of the spatiotemporal variations of these exposures linked to the network density and architecture and the characteristics of the environment. Statistical and geostatistical methods could be used to assess spatiotemporal variations. The approach should allow definition of exposure scenarios adapted to the characteristics of exposed populations and to the ICT usage. Probability methods such as Monte Carlo algorithms could be used to assess the plurality of lifestyles and usages. Finally, these methods should be integrated in an advanced tool that would compromise between time consumption and the complexity of the exposure scenarios.

The AMPERE project (“Advanced MaPping of residential ExposuRE to Rf-Emf sources”) aims at filling this gap by developing a statistical tool for spatiotemporal Rf-Emf exposure mapping by means of advanced statistical methods. Benefiting from previous works in exposure assessment [7,15,16,17], temporal dosimetry [18], geostatistics [8,12,19,20,21], and data fusion [14,22,23,24], the AMPERE project analyses the time, spatial, and statistical characteristics of far-field exposure, i.e., indoor and outdoor exposures and near-field exposure, e.g., exposure from mobile phone or linked to mobile phone usage. These exposures are affected by geographical factors such as land use (density of buildings, outdoor areas) and population characteristics such as age, status (residential, occupational), exposure duration, and device usage. The exposure characterization requires the use of statistical methods such as machine learning, kriging, and stochastic geometry, to build innovative surrogate models of Emf exposure. The aim of this paper is to present the design of an advanced tool for the spatiotemporal Rf-Emf exposure mapping. From surrogate models, the tool combines exposure from outdoor and indoor sources as well as wireless devices. It also includes the characterization of the microenvironments, populations, and their lifestyles, i.e., the time spent in the microenvironments, thus defining exposure scenarios.

## 2. Materials and Methods

Figure 1 summarizes the approach used in this study. This approach combines data characterizing several sources including outdoor base stations, indoor devices and mobile phone devices as well as their spatiotemporal variations (Figure 1). For the characterization of far-field, i.e., downlink exposure, this approach used outdoor electric-field levels interpolated from base stations and indoor SAR levels estimated in a room from indoor devices. Time profiles were constituted to consider the varying usage of telecommunications networks as well as temporal changes in the environment (Figure 1, [25]). For the near-field exposure, i.e., uplink, mobile phone SAR levels were used considering mobile phone usage (Figure 1). Moreover, three microenvironments were defined: outdoor areas, homes, and offices. Population and typology data were used to characterize these microenvironments (Figure 1).

For a fine characterization of the Rf-Emf exposure, a regular grid was used to aggregate information from microenvironments and to combine the associated populations as well as the time they spend in the area. The Paris 14th district was, therefore, subdivided into a grid with a spatial resolution of 100 × 100 m in which each variable of interest was described. A probabilistic approach based on Monte Carlo (MC) algorithms was used to estimate exposure distributions. With this method, uncertainties were propagated throughout the assessment process. MC algorithms also allowed one to consider the variability, both temporal and spatial, of Rf propagation in microenvironments and the distribution of population over the analysis grid (Figure 1). However, the characterization exercise was complicated due to different Rf units. In view of standardization, specific tools were used to convert all metrics into SAR (mW/kg) [26]. By the methods and data used, the procedure was articulated in a tool implemented in a Geographic Information System (GIS; Figure 1).

As part of this study, we designed a statistical tool with a graphical user interface (GUI) that provides automated geospatial data processing to assess and map aggregated exposure to Rf-Emf sources (Figure 2). It integrates data that characterize an analysis area and allows one to define parameters for exposure scenarios. To be useful to entities in charge of the risk assessment, to epidemiologists, and to the general public, the developed toolboxes were designed to be as simple as possible. The tool was designed under the software (version 2.18), which is linked with Python programming language [27]. All geoprocessing QGIS methods necessary to the approach were run on this software and, also on R (version 3.3.2) via the “Processing” framework [27]. The linkage between R, QGIS, and Python provides data processing methods that are not directly available on each software.

The aggregated exposure assessment was conducted on the 14th district of Paris, which is densely urbanized, to illustrate the functionalities of the statistical tool.

### 2.1. Far-Field Exposure

#### 2.1.1. Outdoor Exposure

Different approaches have been adopted to monitor downlink levels, such as simulation models and on-site measurements [4,28]. Measurements give precise and reliable information.

Electric-field measurements were performed on 800, 900, 1800, 2100, and 2600 MHz frequency bands dedicated to downlink transmission used by cellular network operators in the 14th district of Paris. A three-axis wideband antenna was installed on top of a car at 2 m from the ground connected to a spectrum analyzer (NARDA SRM-3006). The average speed of the car during the measurements was 27 km/h. The electric-field values were measured on three axes for each frequency band and GPS coordinates (Global Positioning System) were recorded for each measurement point. The sweep time was fixed to 2.5 s and the time between two measurement points to 5 s. In total, 1100 measurement points were recorded during the trial to cover the entire district.

Each grid cell was 2.4 points on average. To assess the possibility of extracting exposure information from sparse data, we used the full dataset in combination with ordinary kriging. Since downlink exposure was log-normally distributed, a logarithmic transformation was applied to obtain normally distributed data. Ordinary kriging was calculated by considering only interpolation points in a local neighborhood of the point to be predicted, thus obtaining a local varying mean, which has been chosen equal to four times the variogram range. Estimated exposure was then back-transformed into linear scale by applying the following operations (1 and 2) [29,30]:(1)$$\tilde{z}\left(x\right)=e^{\left({\tilde{z}}_{log}\;\left(x\right)+\sigma_{log}^{2}\left(x\right)/2\right)},$$
(2)$$\sigma^{2}\left(x\right)=e^{2\times {\tilde{\;z}}_{log}\;\left(x\right)+\sigma_{log}^{2}\left(x\right)}\times \left(e^{\sigma_{log}^{2}\left(x\right)}-1\right),$$
where $\tilde{z}\left(x\right)$ and $\sigma^{2}\left(x\right)$ are, respectively, the predicted value and the kriging variance at point ***x***, ${\tilde{z}}_{log}\;\left(x\right)$ is the predicted value in natural logarithmic scale, and $\sigma_{log}^{2}\left(x\right)$ is the kriging variance in logarithmic scale.

Leave-one-out cross-validation of the interpolation in the exhaustive dataset was carried out [31]. Besides that, we ran the leave-one-out cross-validation by excluding neighbors of the point to be estimated, in order to analyze the effect of increasingly larger areas without measurement.

#### 2.1.2. Time Variation

The variability due to network activity can be included in the Rf-Emf exposure assessment by assigning to the point of evaluation a temporal profile that encompasses, for example, the average diurnal evolution in network activity [18]. With this temporal profile, the exposure value at the point of evaluation can be scaled based on the time of the measurement and the time of the actual evaluation. In the literature, analyses of temporal variations are rare and mostly use repeated measurements instead of monitoring networks [18].

To this end, temporal Rf-Emf data points were collected in two monitoring networks consisting of several fixed Rf-Emf measurement devices (also called “nodes”). The first network is located in Santander, Spain (SmartSantander, http://maps.smartsantander.eu/) [18,32], contains 36 nodes, and has been active for more than four years; the second network is located in Antwerp, Belgium (Smart Zone, https://antwerpsmartzone.be/en/), contains 10 nodes, and has been active for about one year [33]. Both networks are part of a larger Internet-of-Things (IoT) platform. In both cases, the nodes periodically measure the electric-field strength in three different frequency bands containing downlink telecommunications signals: 925–960, 1805–1880, and 2110–2170 MHz. In the Santander network, the sample rate was once per five or fifteen minutes; in the Antwerp network, the sample rate was once per second.

In order to diminish the influence of long-term variations in the signals, e.g., due to lasting changes in the network or environment, the electric-field measurement samples *E_i_* were first normalized by dividing them by the average field value of the day $\bar{E_{day}}$ (3) [18]:(3)$$\eta_{i}=\frac{E_{i}^{2}}{{\bar{E_{day}}}^{2}}\;.$$

The normalized field values, $\eta_{i}$, were aggregated and averaged per hour of the day to obtain the diurnal temporal profile for each of the measured frequency bands [18]. By combining the measurements from the entire monitoring network, a single profile was obtained for the geographic area covered by that network. This profile encompasses the diurnal variability of the Emf emitted in those bands and can thus be used to scale a measurement performed on one moment of the day to another moment of the day [18].

Time profiles were afterward associated with the outdoor electric-field estimates to assess the temporal variability in this area. First, an initial distribution of 20,000 samples was built for each grid cell, thus constituting a non-time-varying outdoor exposure profile (SciPy module [34], version 0.18.1). We assumed that the distribution is normal and takes the linear back-transformations of outdoor electric-field estimates and the kriging variance as the *µ* and *σ^2^* parameters, respectively. This distribution was then combined and weighted with each of the measured frequency bands’ time profiles. Finally, the temporal outdoor exposure distribution parameters were fitted with both gamma and normal distributions and both fits were compared with the Kolmogorov–Smirnov test [34].

#### 2.1.3. Indoor Exposure

Considering the uncertainty of the environment knowledge, indoor antenna, and user locations, the following described methods were used to build innovative surrogate statistical models of the Emf indoor exposure due to some typical radio frequency (Rf) sources.

The exposure of a child moving in a 3 × 4 m^2^ room to two different sources with uncertain position, a wireless local area network (WLAN) access point operating at 2.4 GHz and a 4G LTE femtocell operating at 2.6 GHz, was assessed by varying their position on the wall and the position of the child [35,36]. The position of the sources was described by the horizontal location and the height, while the position of the child was described by the position on the floor (coordinates x and y) and the rotation along the vertical axis. The exposure was evaluated in terms of whole-body specific absorption rate (SAR), for different positions of the sources and the child using surrogate models based on low-rank tensor approximation (LRA) [37,38]. Each surrogate model describes how the variable of interest *Y*, i.e., the SAR levels, is affected by the variability in the input parameters *X*, i.e., the spatial coordinates of the femtocell and the child.

LRA is a non-intrusive method, in which the phenomenon to approximate is treated as a “black box,” for developing surrogate models as a finite sum of rank-one functions (4).
(4)$$Y_{LRA}=\hat{M}\left(X\right)=\sum_{l=1}^{R}b_{l\;}w_{l}=\sum_{l=1}^{R}b_{l\;}\left(\prod_{i=1}^{M}v_{l}^{\left(i\right)}\left(X_{i}\right)\right),$$
where $w_{l}$ is the *l*-th rank-one function obtained as product of univariate functions of the components of $X_{i}$, $v_{l}^{\left(i\right)}$ denotes a univariate function of the components of $X_{i}$ in the *l*-th rank-one component, M is the number of input variables, $b_{l\;}$ (*l* = 1,…,R) are normalizing constants, and R is the rank of the decomposition. The coefficients of the surrogate exposure model were estimated minimizing the error between *Y* and *Y*_0_, i.e., the SAR level computed at N specific input spaces *X*_0_, called the experimental set. The first step consisted of achieving the *Y*_0_ set by computing the levels of SAR for N positions of each source and child in the room, using computational electromagnetics methods. Starting from these exposure values, the LRA procedure was applied to develop a proper surrogate model. Finally, the obtained surrogate model was used in child exposure analysis [35,36].

In order to obtain the two-dimensional (2D) spatial distribution of the electric-field (E) induced in a realistic apartment by a Wi-Fi source placed in unknown location, a recently proposed stochastic method combining principal component analysis (PCA) and kriging method was used [39]. The set of observations needed to develop the 2D surrogate models of the exposure was obtained by the WiCa Heuristic Indoor Propagation Prediction (WHIPP) tool, a set of heuristic planning algorithms developed for network planning in indoor environments [40]. First, a kernel PCA with linear kernel was applied. The reason for using PCA is that the E induced at nearby spatial coordinates could be hypothesized to be highly correlated, and thus can be efficiently represented by a few *d* components. As a second step, the kriging method was applied to develop a separate surrogate model for each of the *d* (non-physical) components identified by PCA. The third step in the 2D surrogate modeling procedure consisted of using inverse PCA to reconstruct, from the univariate surrogate models obtained by kriging, the 2D spatial distribution of E in the apartment. Finally, the so-obtained 2D surrogate model was used in analysis of the exposure levels in the apartment [41,42].

Indoor exposure also depends on radiofrequency fields emitted by outdoor sources, which penetrate buildings while being attenuated by the construction materials, the distance, and the type of surrounding buildings [43,44]. Indoor electric-field, after building penetration loss, is approximated by (5),
(5)$$V_{I}=V_{O}\times 10^{-dB/20}.$$

Since the outdoor estimates are in 900 MHz equivalent volts per meter, the building penetration loss is equal to 9.5 dB, corresponding to the mean value observed in the literature for this frequency band [45,46,47,48,49,50,51,52,53,54,55,56,57].

### 2.2. Near-Field Exposure

Mobile phone exposure is cumulative with outdoor and indoor exposures. This exposure reflects two main usages of mobile phones, i.e., voice calls and data transfers via base stations or Wi-Fi (Figure 1). Other parameters affect the exposure such as call time, laterality, and posture for voice mode [11,58,59], volume of data traffic for data mode [11], and uplink transmitted power (${\bar{P}}_{x}$). For both modes, physiological data were also used (body weight, head weight, age [60]). Some of these parameters were calibrated with Mobi-Expo data [9,10]. ${\bar{P}}_{x}$ values were measured in the 14th district through Nemo Handy trace mobile phones while uploading 100 MB files on an FTP (File Transfer Protocol) server repeatedly.

Whole-brain and whole-body SAR transfer functions were computed by using equations from the RF-IEM (RF-EMF Integration Exposure Model) tool developed within Geronimo and CREST (RF exposure induced by novel use and new technologies from mobile communication systems) projects and exposure was estimated with the equations from the LEXNET study that are providing the whole-body and whole-brain SAR per unit of time and W/m^2^ for downlink exposure and per watt emitted from the devices [11,27] (6).
(6)$$E_{tel}=\frac{0.25\times p_{3G}}{24\times 3600}\times t\times SAR_{v}\times p_{v}\times {\bar{P}}_{v}+\frac{0.25}{24\times 3600}\times p_{3G}\times V_{d}\times SAR_{d}\times p_{d}\times \frac{{\bar{P}}_{d}}{\bar{T}},$$
where $SAR_{x}$ are the reference SAR values for voice and data modes, $p_{x}$ are the percentages of 3G connections, voice call users and data traffic users, ${\bar{P}}_{x}$ are the transmitted power values for voice and data modes, t is the duration of voice calls, $V_{d}$ is the volume of data traffic, and $\bar{T}$ is the mobile network connection speed.

Transfer function distribution parameters for voice and data were computed using reference SAR values reflecting the age of the individual and a specific usage. For exposure calculation, the SAR transfer function distributions were weighted with the uplink transmitted power (${\bar{P}}_{x}$).

### 2.3. Population Data Geoprocessing

To estimate populational exposure, it was necessary to build a metric based on the population interacting with the area and space–time budgets, i.e., person-hours. The use of an analysis grid allowed the aggregation of population data, assessed on finer administrative grids.

#### 2.3.1. Exposure Scenarios

We considered different population categories depending on exposure pathways. For the indoor exposure, we considered two population categories: residential and occupational. We grouped the residential population into two age groups: children (less than 15 years) and adults (more than 15 years). The occupational population was made up of a sole age group (from 15 to 64 years). For the outdoor exposure, we considered two populations, mixing residential and occupational populations, and related to the age groups defined beforehand.

Behavioral differences in mobile phone use across age or population groups were not considered. Similar to the Wi-Fi use, we considered that different usages (Skype video call, YouTube video, surfing a new website) have the same proportions.

#### 2.3.2. Assessment of Population in Buildings (Residents and Workers)

The first step in the calculation procedure is the estimate of the population in buildings according to their employment, i.e., resident or occupational activity. A building typology was constructed using data from the MAJIC (Cadastral Information Update) database (Cerema, https://datafoncier.cerema.fr/donnees/fichiers-fonciers) to locate buildings and define their function.

The population in buildings, named $P_{i}$ was estimated for each age class and category by a surface ratio using IRIS (Grouped Blocks for Statistical Information) data from the 2015 national census (National Institute of Statistics and Economic Studies, INSEE) (7):(7)$$P_{i}=\sum_{j=1}^{n}\frac{S_{ij}}{S_{j}}P_{j},$$
where $S_{ij}$ is the housing area intersecting the IRIS, $P_{j}$ the residential population in the IRIS, and $S_{j}$ the total housing area in the IRIS. Similarly, the occupational populational was assessed using aggregated census data from the 14th district.

#### 2.3.3. Person-Hours Estimation

Person-hours were estimated for each population category as the product of the number of people and the time spent in their respective microenvironment (indoor, outdoor, and mobile phone use). Data used to estimate time spent were derived from French national studies: a 2008 mobility campaign, a 2010 study on indoor air quality, and a 2015 individual working hours census [61,62,63].

### 2.4. Aggregated Exposure Assessment

Aggregated exposure was assessed for the whole 14th district. Exposure distributions were calculated for each grid cell and building using MC algorithms and the distribution parameters that were computed for each exposure source. Person-hours values were used to define the number of samples in the exposure distributions. There were as many distributions such as population categories and organ tissues considered. Grid cell distributions combined time-varying outdoor and mobile phone exposure distributions. Similarly, building distributions combined all indoor sources (WLAN, femtocell, Wi-Fi use), outdoor-to-indoor attenuation, and mobile phone exposure distributions.

On each grid cell, the aggregated outdoor exposure distributions were combined with the aggregated indoor exposure distributions of all the buildings intersecting the grid cell. The final exposure distributions were then segregated into 20 percentiles, using the NumPy module (version 1.11.3, https://numpy.org/, [64]). It is important to note that all aggregated exposure estimates were given in SAR (mW/kg). As some exposure distribution parameters, especially for outdoor and Wi-Fi distributions, were computed in volt per meter (V/m), these distributions were converted into SAR using the RF-IEM tool [26].

## 3. Results

### 3.1. Far-Field Exposure

#### 3.1.1. Outdoor Exposure

The estimates of outdoor electric-field values are shown in Figure 3a. Estimates are spatially heterogeneous with higher values in the south of Paris 14th district, near the beltway. Two hotspots are located in the northwest and central northeast of the district (surroundings of Montparnasse station and Denfert-Rochereau square). With the consideration of temporal variations, the outdoor median exposure tends to be slightly lowered over most of the 14th district. Nevertheless, the median exposure increases more sharply in the northwest and the south near the beltway (Figure 3b).

#### 3.1.2. Indoor Exposure

SAR values for WLAN and femtocell combined exposure profiles are exceeding Wi-Fi use exposure profiles by a thousand-fold factor. WLAN and femtocell SAR values are higher for the whole body than for the whole brain with a two- to fourfold factor (Table 1). Conversely, Wi-Fi whole-brain SAR values are slightly higher than those for the whole body, by 35% for adults and 15% for children. Meanwhile, child SAR values are also slightly higher than those of adults, exceeding by 22% for whole body and 4% for whole brain. All distributions are strongly spread, with coefficients of variation (i.e., the ratio of the mean over the standard deviation) ranging between 0.68 (WLAN + femtocell profile for whole brain) and 1.17 (Wi-Fi profile for adult whole body) (Table 1).

Outdoor electric-fields contribute to indoor exposure but are of course attenuated by buildings, walls, and windows. However, estimates are low and, globally, do not reach 0.4 V/m in most part of the 14th district. Two hotspots remain near Denfert-Rochereau square and Montparnasse station where a peak of 0.82 V/m has been estimated (Figure 4).

### 3.2. Mobile Phone Exposure

SAR values of adult and child mobile phone exposure profiles display opposite trends depending on the tissue. Children have whole-body SAR values from 50% to 80% higher than those of adults. Inversely, adults show whole-brain SAR values 50% higher than those of children (Table 2).

### 3.3. Aggregated Exposure Assessment

Figure 5 shows the results of the aggregated exposure assessment, averaged over the whole population and the time spent in the area. On the left side, whole-body maps of the median percentile, mean, and standard deviation are displayed. On the right side, the same results are displayed for the whole brain. As expected, whole-body SAR estimates globally exceed those of the whole brain with a 2.7-fold factor. On the median exposure maps, whether for the whole body or the whole brain, some areas located in the northwest, east, and south of the district display lower values. These areas correspond to open spaces such as parks and the beltway. These areas also display lower mean and standard deviation values, which show that the spatial dichotomy between open spaces and built-up areas is maintained across all percentiles (Figure 5c–f).

Figure 6 shows that indoor exposure is the main contributor to aggregated exposure for both whole body and whole brain, followed by outdoor exposure and mobile phone exposure. The outdoor exposure contribution increases from the lower percentiles to the upper percentiles. It reaches 3.16% on the 99th percentile for the whole body and 43.71% for the whole brain. Mobile phone exposure can also be considered negligible as it represents globally less than 1% of the aggregated SAR values. Its contribution increases regularly and reaches a peak of 0.43 (whole-body) 1.18% (whole brain) on the 80th percentile before decreasing. This result is in contrast with two previous studies that establish the mobile phone contribution to aggregated exposure at 32% and 73%, respectively [24,65]. Unlike our study, these efforts have analyzed GSM exposure, this source being at the time the largest contributor to exposure [24,65]. We did not integrate this source as GSM is an old technology gradually replaced by 3G, 4G, and 5G [1].

## 4. Discussion

In this study, we developed an innovative approach to assess aggregated populational exposure to Rf-Emf. The approach highlights the complexity of this exposure, i.e., the multiplicity of sources, the usage of wireless devices as well as geographic factors, i.e., microenvironments. This complexity not only leads to spatial and temporal variability of the exposure but also generates uncertainties due to the difficulty in characterizing exposure [25]. To guarantee the inclusion of the variability aspect and the propagation of uncertainties, we used a probabilistic approach based on Monte Carlo (MC) algorithms. Unlike the deterministic approach, this allowed the establishment of exposure profiles, aggregating the population characteristics as well as their space–time budgets. Furthermore, several studies have shown exposure inequalities according to the location of populations and environments, which confirms the spatial dimension of Rf-Emf exposure [66,67,68,69,70,71]. For spatial characterization of exposure, we therefore integrated geographic information and exposure data. These data required specific processing methods, and we used a GIS environment to combine both geoprocessing and MC algorithms. Additionally, we have developed an aggregated exposure assessment platform that tends to be as simple as possible and does not require too large expertise in telecommunications, propagation, and electromagnetism. This tool is able to integrate different types of data and to define all of the data processing stages. By mapping the Rf-Emf population exposure, the platform provides relevant and easily comprehensible information for the public, health researchers, and policy-makers.

In this case study, resulting aggregated exposure maps display a dichotomy between outdoor areas and built-up areas. The latter areas reach higher SAR values as indoor exposure is more prevalent than outdoor exposure. This result is of important concern as people spend almost 90% of their time indoors [72]. Maximum aggregated exposure SAR for the whole body and the whole brain at the 99th percentile is respectively estimated at 3.5 and 3.9 mW/kg. These estimates are significantly below the recommendations of the International Commission on Non-Ionizing Radiation Protection (ICNIRP) that are referenced at 80 and 2000 mW/kg for these tissues [73]. Nevertheless, the exposure assessment still contains gaps and uncertainties: some exposure sources were not quantified such as exposure to 4G LTE technology. The RF-IEM tool, used for mobile phone exposure assessment, does not yet predict SAR values for this technology as it lacks information regarding output powers and data to characterize the source. Future investigations will enable the platform to assess 4G exposure and will also have to integrate the emerging 5G technology. A second significant gap concerns the lack of substantial mobility data, both in terms of localization of mobility and the number of people transiting through the area. Thus, assumptions we made may lead to an underestimation of the outdoor exposure contribution. A solution could be found by using smartphone signals in tracking techniques to locate people outdoor [74].

Median outdoor exposure estimates have an average of 0.66 V/m (Figure 3b). That is almost twice the average of Flemish cities (0.37 V/m) and Melbourne’s central urban areas (0.35 V/m) [75,76]. This difference can be explained by the population density in the 14th district, which is around 24,000 inhabitants per square kilometer, while densities for these other cities are generally below 2500 inhabitants per square kilometer. Inner Brussels and Melbourne, which are more densely populous, display similar median outdoor estimates, each one reaching 0.59 V/m [75,76]. Outdoor exposure was quantified by an ordinary kriging method, and the quality of the interpolation was cross-validated by leave-one-out method. In a different approach, Isselmou et al. used kriging with external drift. They combined measurement data with a geostatistical model output as the external drift [7]. In our case, the geostatistical model could be improved with Cartoradio measurement data (https://www.cartoradio.fr/index.html). Measurements would also serve as control points for the goodness of interpolation. Meanwhile, outdoor exposure time-variation was assessed from dedicated Rf-Emf sensor networks with multiple sensor nodes in relative proximity (100 nodes per km^2^) which, to our knowledge, are only found in Santander and Antwerp [18,32,33]. In both cases, the areas covered by the Rf-Emf sensors are found in the city centers, and they were considered surrogates for Paris 14th district for which no temporal Rf-Emf data were available. However, these data came from smart cities that potentially had a higher exposure level given the large number of smart sensors that are deployed [77]. Thus, time profiles can represent a conservative approach of outdoor exposure.

Regarding indoor exposure, estimates are consistent with previous studies [35,36,42]. However, some uncertainties remain. On the one hand, WLAN and femtocell distribution parameters used for the assessment of adult indoor exposure were actually estimated on children. Using children’s distribution parameters to assess adult indoor exposure can be considered as a conservative approach since children’s SAR values are higher than those for adults [73]. Moreover, indoor exposure was only assessed from modeling and not adjusted with existing measurement data. For a better characterization, Cartoradio indoor measurement data will be integrated to spatially correlate model outputs with measurements and rescale estimates using geostatistical methods such as kriging with external drift. The larger contribution of mobile phones compared to indoor sources in previous studies can reflect the lower exposure from 3G compared to that from 2G mobile phones. It could again be the case with 4G phones that the phone contribution comes back to be larger than in this study. On the other hand, we have made simplified modeling assumptions regarding the building penetration loss. For a realistic estimate, one should consider the influence of building materials in Rf-Emf propagation instead of fixing a deterministic value. However, if several building penetration loss values are found in the literature, there is little knowledge on the impacts of building materials in these attenuation mechanisms [55,78]. Finally, uncertainties remain about the contribution of other indoor sources coming from adjacent apartments or premises.

Given the rapid growth and evolution of ICT, the exposure assessment is also depending on data availability and age. The Mobi-Expo study took place between 2012 and 2014, and the resulting data are getting older since mobile phone usage has changed since then. Mobility data came from a 2008 campaign and since, mobility habits may have changed with the advent of new transportation modes and homeworking. The next campaign, covering the years 2018 and 2019, will not be available sometime before 2020. Moreover, there is a lag of 3 years between population census and public availability of data. The lag between data acquisition and availability alongside the disuse of these data could mean our mapping of the Rf-Emf exposure is slightly outdated already and may not reflect current populational exposure.

Despite the gaps and uncertainties surrounding the exposure assessment, the platform provides relatively low computation times, as it rapidly processed data and produced aggregated exposure maps in less than one hour for the 14th district. This case study gives encouraging results to run this platform on larger territories and other frequency bands, i.e., low-frequency or medium-frequency bands. The approach and the geoprocessing methods are easily reusable, and we could consider adapting this platform for exposure assessment to chemical agents or biological risk factors, notably to map environmental inequalities.

## 5. Conclusions

We have designed a statistical tool that uses surrogate models for spatiotemporal mapping of Rf-Emf exposure. Using a mathematical and probabilistic approach, this tool aggregates spatiotemporal surrogate models of Rf-Emf exposure with geographical data, population distributions, socioeconomic data, and ICT use patterns. Data fusion of all this heterogeneous information results in an innovative and advanced mapping of residential exposure providing information on variability, occurrence, and exposure quantiles relevant for the public, health researchers, political decision-makers, and risk assessors. Future works will improve the approaches developed. For the Rf-Emf exposure, some exposure sources (laptop, 4G, 5G, and other future technology) will have to be integrated into the exposure assessment to consider the evolution of technologies and usages. Certain exposure mechanisms such as outdoor-to-indoor attenuation or electric-field propagation will need to be better characterized. For the methodological aspect, some methods used in this study will also be applied to larger and heterogeneous territories as well as for exposure assessment to chemical agents or biological risk factors.

## Figures and Tables

**Figure 1 ijerph-17-05339-f001:**
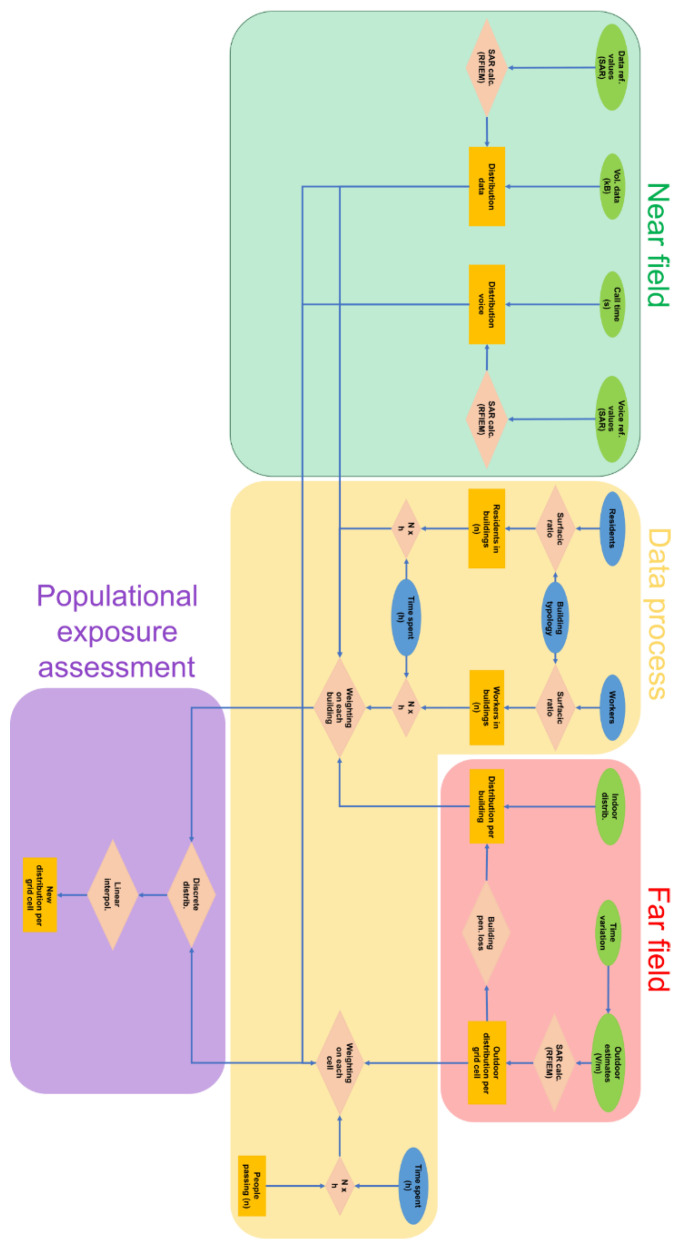
Exposure assessment framework. Colored boxes depict methodological processes that are described hereafter: far-field exposure assessment in red, near-field exposure assessment in green, data fusion of geospatial data in yellow, and populational exposure assessment in purple.

**Figure 2 ijerph-17-05339-f002:**
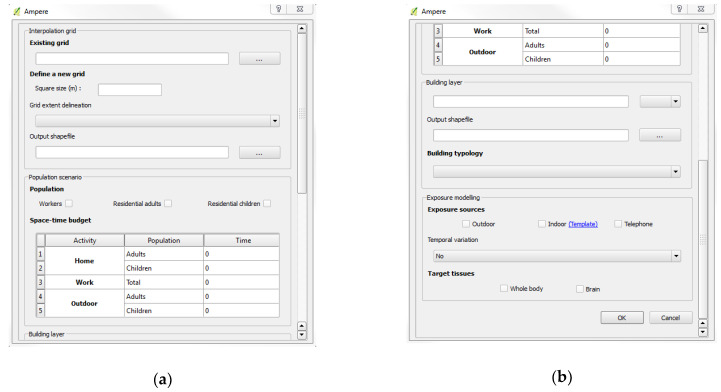
Screenshots of the graphical user interface (main toolbox) developed in the plugin: (**a**) Top of the main toolbox; (**b**) bottom of the main toolbox.

**Figure 3 ijerph-17-05339-f003:**
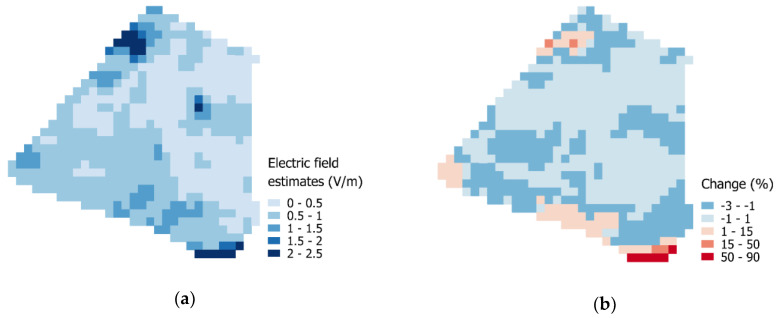
Outdoor exposure estimates: (**a**) electric-field estimates from ordinary kriging (V/m); (**b**) percentage change in median exposure after taking time profiles into account.

**Figure 4 ijerph-17-05339-f004:**
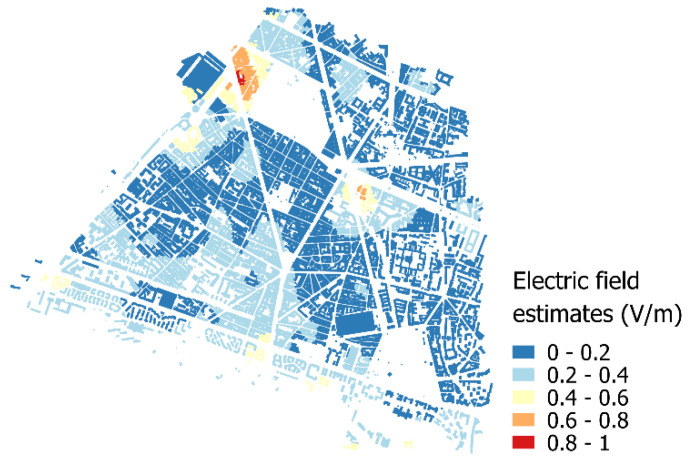
Estimates of outdoor electric-field strength (V/m) in buildings after building penetration loss.

**Figure 5 ijerph-17-05339-f005:**
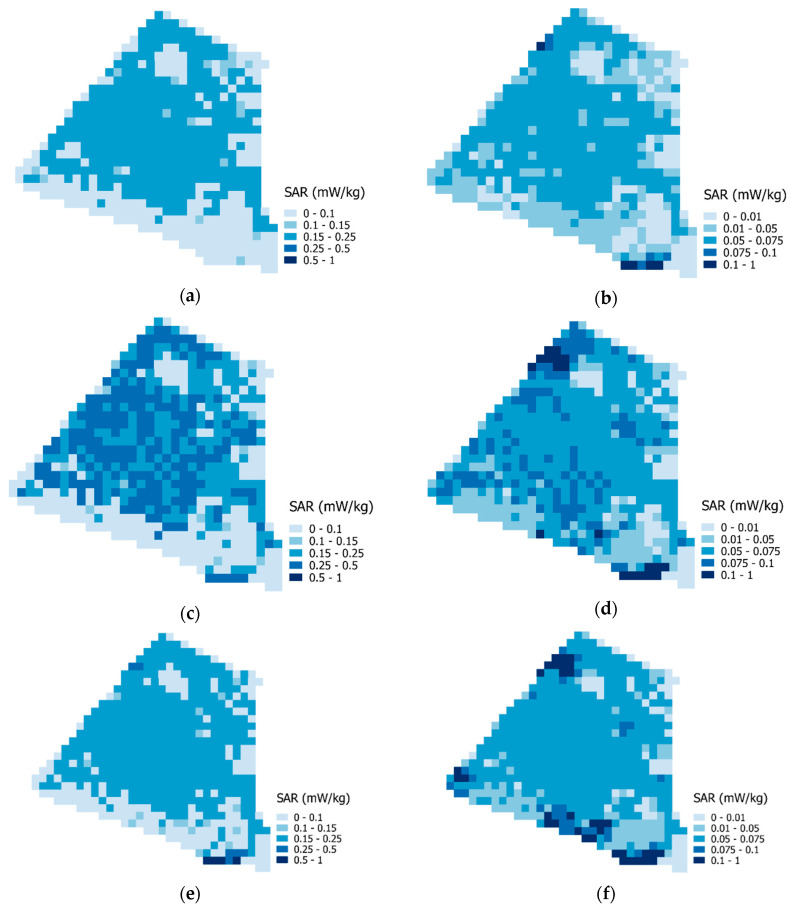
Radiofrequencies aggregated exposure maps (in SAR): (**a**) whole-body median percentile; (**b**) whole-brain median percentile; (**c**) whole-body mean; (**d**) whole-brain mean; (**e**) whole-body standard deviation; (**f**) whole-brain standard deviation. Each map portrays a statistical indicator that aggregates the exposure of all the people passing through the study area, weighted by the time they spend in there. For example, a value of 0.1 mW/kg in the top left panel indicates that it is the median of the whole-body exposure of all people that spend at least some time in that cell, weighted by the time they spend in there.

**Figure 6 ijerph-17-05339-f006:**
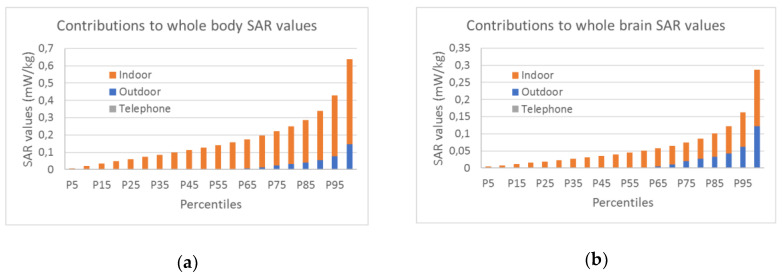
Estimated contributions to SAR values: (**a**) whole body; (**b**) whole brain.

**Table 1 ijerph-17-05339-t001:** Descriptive statistics of the cumulative distribution functions of indoor exposure profiles (specific absorption rate (SAR) in mW/kg).

Indoor Source	Tissue	P25	P50	P75	Mean	Standard Deviation
**WLAN + Femtocell**	Whole Body	6.9 × 10^−2^	1.7 × 10^−2^	3.4 × 10^−1^	2.4 × 10^−1^	1.7 × 10^−1^
	Whole Brain	3.2 × 10^−2^	5.6 × 10^−2^	9.1 × 10^−2^	6.7 × 10^−2^	4.9 × 10^−2^
**Wi-Fi**	Adult Whole Body	8.2 × 10^−6^	2.1 × 10^−5^	4.8 × 10^−5^	3.6 × 10^−5^	4.2 × 10^−5^
	Child Whole Body	10 × 10^−6^	2.6 × 10^−5^	5.9 × 10^−5^	4.3 × 10^−5^	5.1 × 10^−5^
	Adult Whole Brain	1.1 × 10^−5^	2.9 × 10^−5^	6.6 × 10^−5^	4.9 × 10^−5^	5.7 × 10^−5^
	Child Whole Brain	1.2 × 10^−5^	3 × 10^−5^	6.8 × 10^−5^	5 × 10^−5^	5.9 × 10^−5^

**Table 2 ijerph-17-05339-t002:** Descriptive statistics of the cumulative distribution functions of mobile phone use exposure profiles (SAR in mW/kg).

Tissue	Population	P25	P50	P75	Mean	Standard Deviation
**Whole Body**	Adult	1.1 × 10^−4^	2 × 10^−3^	1.6 × 10^−2^	6.3 × 10^−3^	3.1 × 10^−2^
	Child	4.5 × 10^−5^	8.3 × 10^−4^	6.8 × 10^−-3^	3.5 × 10^−3^	1.5 × 10^−1^
**Whole Brain**	Adult	6.8 × 10^−5^	1.3 × 10^−3^	1 × 10^−2^	3.6 × 10^−3^	1.4 × 10^−2^
	Child	1.6 × 10^−5^	2.9 × 10^−4^	2.4 × 10^−3^	8.4 × 10^−4^	1.7 × 10^−3^

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
