# Peer review of "Design of an Integrated Platform for Mapping Residential Exposure to Rf-Emf Sources"

_ijerph, 2020, doi:10.3390/ijerph17155339_

Round 1

Reviewer 1 Report

The authors should clarify in the abstract if the word “respectively” in lines 30-32: “Maximum values do not exceed 3.5 and 3.9 mW/kg respectively, thus significantly below the ICNIRP recommendations”, relates to the whole body SAR or whole brain values, or to the built-up and non-built-up areas.

In this paper, a statistical tool for RF-Emf populational exposure assessment and mapping is presented. It integrates geographic databases and surrogate models to characterize spatio-temporal exposure from all surrounding sources. A case study to illustrate the functionalities of this tool, is also presented. In the discussion section of the paper, the authors present thoroughly certain gaps and uncertainties in their approach. But, since the subject of the paper is the presentation of this new approach and platform for mapping residential exposure to RF-EMF sources, the authors should give more details on the presentation of their new approach by expanding the Material and Methods section of the paper, e.g. by describing more and justifying the way they performed the on-site measurements in 2.1.1 and how and why they chose the presented “nodes” in 2.1.2 and not other bigger similar networks.

Reference 42 should not be mentioned since it is a paper under review.

Reference 73 should be corrected with the published ICNIRP 2020 guidelines.

The reference stated in line 422 is not written.

Reviewer 2 Report

Thank you to all authors for their work and time. Very important and current topics. I rate the level of manuscript and research carried out by your team very highly. I have only a few editorial comments.

Line 127: The word "aggregate" appears twice closely together. Maybe it is worth changing it if it is possible.

Figure 1: In my opinion, the quality of this drawing is poor. This drawing is unreadable. Scalable vector graphics should be used here.

Figure 2: (a) and (b) under figure are in a different line.

Equation (3): is the denominator ok? There is a small square

Line 262: words Figure 1 should be together

Figure 6. The same like in figure 2

Line 422: Reference error

Reviewer 3 Report

Th exponential growth of communication systems, especially with the arrival of 5G.forces us to have reliable evaluation procedures that allow us to determine the emf inmission rate that the population can support. for this, probabilistic tools such as those presented can be useful, if they are adjusted to the reality of the population exposure rate. Therefore, it is very important to adjust the starting data to compare the values obtained with reality. Thus you have only used data acquisition procedures in measurements outside the home or workingplaces, and have not measuremed internal exposure. Therefore, the data used are only probabilistic,and do not respond to the area evaluated by  the 14th district of Paris.

On the other hand, the procedure for acquiring data on an average basis prevents knowing the contribution of each of the frequencies in the impact on the population. I believe that the measurements should be carried out indoors and outdoors, with "peak to peak" spectrum analysis. What to allow knowing the real impact.

The number of sources evaluated does no correspond to the entire spectrum, not only the new frequencies corresponding to 5G, but the current ones of the wifi systems etc, or of different frequencies have not be suitable evaluate the impact of the whole on the population, at least we do not have this part.

it is important to know with this model if the proximity of the radiating systems contributes to the modification of the data of the models, new vehicle control system will close the antennas.

The probabilistic model should be adjusted to the maximum for the area and not supress sources of exposure for simplicity.

Round 2

Reviewer 3 Report

Dear doctor.
Sorry for the delay in the response but the covid 19 has not left me much capacity to work.
After reading the authors' response, I consider:
1- the article can be published if peak-to-peak and not just average modeling systems are included.
2-it is necessary to complete the adjustment data with real measurements made inside the houses to check the validity. since the variability of expedition data is too great and cannot be compared to averaging process.
3- this tool must have the capacity to expand to new frequencies of the new 5G
If this data is completed, I think it may be a candidate for publication.